# A Trustable and Secure Usage-Based Insurance Policy Auction Mechanism and Platform Using Blockchain and Smart Contract Technologies

**DOI:** 10.3390/s23146482

**Published:** 2023-07-18

**Authors:** Wen-Yao Lin, Kuang-Yen Tai, Frank Yeong-Sung Lin

**Affiliations:** Department of Information Management, National Taiwan University, Taipei 10617, Taiwan; linwy1399@gmail.com (W.-Y.L.); yeongsunglin@gmail.com (F.Y.-S.L.)

**Keywords:** usage-based insurance, blockchain, auction mechanism, internet of vehicles, security

## Abstract

This study presents an architectural framework for the blockchain-based usage-based insurance (UBI) policy auction mechanism in the internet of vehicles (IoV) applications. The main objective of this study is to analyze and design the specific blockchain architecture and management considerations for the UBI environment. An auction mechanism is developed for the UBI blockchain platform to enhance consumer trust. The study identifies correlations between driving behaviors and associated risks to determine a driver’s score. A decentralized bidding algorithm is proposed and implemented on a blockchain platform using elliptic curve cryptography and first-price sealed-bid auctions. Additionally, the model incorporates intelligent contract functionality to prevent unauthorized modifications and ensure that insurance prices align with the prevailing market value. An experimental study evaluates the system’s efficacy by expanding the participant pool in the bidding process to identify the winning bidder and is investigated under scenarios where varying numbers of insurance companies submit bids. The experimental results demonstrate that as the number of insurance companies increases exponentially, the temporal overhead incurred by the system exhibits only marginal growth. Moreover, the allocation of bids is accomplished within a significantly abbreviated timeframe. These findings provide evidence that supports the efficiency of the proposed algorithm.

## 1. Introduction

The internet of vehicles (IoV) is a cloud-based vehicle operation information platform that provides big data and information services from multiple sources. Comprehensive protection measures are required to ensure the security of IoV information [1]. To promote the technological development of self-driving smart vehicles, IoV communication technology and the IoV policies of various countries are both primarily focused on increasing demand for IoV [2,3]. Car dealers and chip manufacturers have adopted strategic alliances and optimized their technical capabilities to capture a larger market territory [4].

Usage-based insurance (UBI) is a type of automobile insurance that leverages driving technology to determine a car’s status by analyzing driving data to calculate insurance premiums on the basis of driving behavior [5] instead of the owner’s age, gender, or accident record. However, conventional UBI car insurance has some disadvantages. The vehicle must be equipped with an onboard diagnostic device (OBD) that monitors and collects driving data. Nonetheless, a potential concern arises in relation to the susceptibility of the device employed in this context, the OBD system, to tampering by the vehicle owner through software modifications. Consequently, the data collected from OBDs may encounter skepticism from authorities [6]. In an effort to address this issue, the present study has developed a blockchain-based UBI platform. By leveraging blockchain technology, the platform facilitates the publication of digitized driving data on the chain in an encrypted or otherwise verified manner, while simultaneously safeguarding personal privacy. Moreover, this approach ensures the credibility of the data and generates value by creating readable and trustworthy intangible assets. As the rightful possessor of these assets, the car owner can exercise unrestricted control over them, such as by providing the corresponding data to insurance or financial organizations in order to obtain individually tailored insurance rates.

After receiving owner authorization, the IoV blockchain platform encrypts vehicle data—such as driving time, period, speed, mileage, and braking—and uploads them to the chain. After the owner’s consent is obtained, other operators on the chain can create customized financial services on the basis of these data [7,8]. For example, car owners with safer driving habits could obtain more favorable car insurance premiums.

However, it is important to note that the IoV blockchain financial platform is currently in the proof-of-concept stage [9,10]. The advent of the blockchain ecosystem is anticipated to drive a new wave of car owners who recognize the fundamental principles of “data belonging to users” and “data as an asset” [11]. This paradigm shift implies that consumers, particularly car owners operating within the blockchain framework, will reevaluate their perspectives on personal information and privacy. For enterprises, the expanded participation within the ecosystem will enable the provision of a wide range of supplementary services for car owners [12]. The utilization of blockchain technology ensures the protection of driver privacy, while the integration of automated UBI technology enhances profitability for insurers. Moreover, the platform proposed in this study has the potential to facilitate reliable and transparent pricing mechanisms for insurers, resulting in benefits for customers. Additionally, the platform could aid automobile manufacturers in refining their products by granting them access to extensive datasets derived from driving activities.

To overcome the difficulties of IoV blockchain development [13,14], the proposed platform connects insurance and financial services on the basis of the driving data of the car owners on the chain. Furthermore, the driving behavior model [15] was used to develop the following application scenarios and address the research goals:(1)Develop a blockchain-based UBI maintenance system: Driving data and insurance claims cases could be compared to identify correlations between various driving behaviors and risks such that insurance plans corresponding to a driver’s typical driving habits can be recommended.(2)Auction mechanism: The implementation of an auction mechanism that is appropriate for the blockchain-based UBI platform has the potential to reduce information asymmetry between the insurance company and consumers, which could result in an increase in the level of trust that consumers have in the platform.(3)Safety reminders: Drivers could be informed of the risk for various road sections while driving; during GPS navigation, safer road sections could be preemptively recommended to the driver. Various application scenarios could be developed from the driving behavior model based on the driving data of the car owners on the chain. Because the IoV enables all machines to communicate with each other, many novel forms of assistance could also be achieved. For example, the car could signal a garage if it notices an abnormal tire pressure sensor reading and remind the driver to check and repair the tire to prevent an accident. The car owner could also directly connect to the system of a financial institution through an application programming interface (API) to have the cost of the repair reimbursed.

## 2. Literature Review

### 2.1. IoV Infrastructure with Blockchain

In recent years, the financial industry has poured resources into blockchain technology. However, achieving its goals regarding this technology, implementing it in business, and creating fundamental scientific or technological financial services have been challenging [16]. A blockchain financial platform for vehicle networking could collect driving behavior, cryptographically store these data, and transmit the data to the blockchain in real time [17,18]. Because the blockchain is incorruptible, companies can link insurance and financial services to authentic and reliable driving information on the chain. Providing car owners with faster and more convenient financing applications, insurance claims, and personalized product recommendations could significantly improve internal enterprise operations and the customer experience [19]. For example, the Data Hub platform built by the electric car manufacturer Tesla regularly collects data from the driving computer of each Tesla car and also has an official open API [20]. Tesla’s blockchain team wrote its own decentralized application (DApp) and smart contracts to establish nodes through charging stations. After the owner is authorized, the DApp directly connects to the official API, and the electric vehicle is designated. The trip computer data are then input to the DApp, and the DApp uploads the data from the node to the chain on the basis of the smart contract [21].

To create financial insurance products that meet individual needs, information from various sources is required, such as driving habits, vehicle information, and consumption habits [22]. Auditing the authenticity of these data is expensive and limits their usage in the business process. For example, to confirm the authenticity of accounts receivable, a bank must also ensure that the statements of both parties and stakeholders are genuine [23]. High audit costs have led to numerous incidents of fraud; however, companies that require financing turnover have enormous operating pressures during information review periods.

Many blockchain developers have shifted their focus to the decentralized finance field by using the incorruptibility of the blockchain to achieve code enforcement (known as Code is Law). This method can greatly reduce the cost of some financial institution activities, such as deposits and lending [24].

### 2.2. Development of UBI System

In UBI, driving data, such as vehicle usage and driving behavior, are analyzed in a cloud system [25]. UBI systems can determine driving risk more accurately than can conventional car insurance systems, which base risk on gender, age, and accident frequency [26]. One advantages of UBI is its high pricing flexibility [27]. For example, drivers who rarely use the accelerator and brake, have safe driving habits, obey traffic rules, and rarely drive during peak hours could have greater flexibility and discounts on insurance costs in a UBI scheme than for conventional auto insurance.

UBI typically requires an OBD to be installed in the car or a mobile app to record driving behavior [28]. Moreover, UBI could also use car maintenance records in addition to owner driving habits to more objectively price car insurance [29]. However, an urgent problem that is encountered when implementing UBI is determining how the owner’s vehicle information and personal data can be obtained. An owner who values their privacy may be concerned about whether personal driving data will be adequately and safely stored; these doubts regarding privacy are a key barrier to the universal implementation of UBI [30,31].

### 2.3. Blockchain-Based Security Auction Algorithm

E-auctions are widely practiced in e-commerce, enabling bidders to participate directly in online product bidding processes [32]. However, in the case of sealed bids, an intermediary must be involved. They were incurring additional transaction costs to facilitate trade between buyers and sellers during the auction. Regrettably, there exists a need for more assurance regarding the trustworthiness of this third-party entity. Blockchain technology can be leveraged to tackle this predicament, offering the advantages of low transaction costs and the ability to establish smart contracts for both public and sealed bids. A blockchain is an immutable and distributed database that maintains transactional data within a peer-to-peer network [33]. The database operates decentralized to allow any member node in the overlay blockchain network to participate in its maintenance anonymously. Transactions between member nodes are recorded in cryptographic hash-linked data structures called blocks. Smart contracts are crucial in ensuring transactions’ security, privacy, irrefutability, and immutability. All transactions are recorded in identical but decentralized ledgers. A smart contract encompasses essential details, such as the auctioneer’s address, auction start time, deadline, the current winner’s address, and the highest bid at the present moment [34].

Blockchain has garnered considerable popularity as an emerging decentralized and secure data management platform [35]. In proof-of-work-based consensus protocols, network nodes, referred to as miners, are motivated to verify new transaction blocks by solving a cryptographic puzzle through a process known as “block mining”. This study primarily focuses on utilizing a blockchain network as an infrastructure for decentralized data management applications, considering the constraints imposed by computing resources. Allocative externalities take precedence when devising auction mechanisms for such a network.

The permissionless blockchain represents an exemplary decentralized and autonomous platform for data management in numerous applications, as it operates without requiring prior authorization [36]. Permissionless blockchains offer the advantage of rapidly establishing a self-organized data management platform capable of supporting a broad array of decentralized applications. Without the need for assistance or permission from trusted intermediaries, individuals can independently design smart contracts and freely develop decentralized applications.

### 2.4. UBI Maintance System Using IoV and Blockchain

The growing interest in leveraging the decentralized nature of blockchain technology has sparked recent research efforts to develop authentication and authorization mechanisms for the IoV [30,37]. These mechanisms seek to eliminate the reliance on centralized third-party intermediaries. This study presents a comprehensive framework that facilitates decentralized and a privacy-centric data exchange in the IoV, explicitly focusing on monetized services. The framework is built upon a tiered blockchain architecture and employs the InterPlanetary file system for secure data storage and transfer [38]. Its main objective is to empower IoV data users to have fine-grained control over data sharing with entities authenticated by the blockchain. Two use cases are provided to demonstrate the practical application of the framework: an IoV data marketplace and decentralized connected vehicle insurance. These examples showcase the framework’s versatility in enabling smart contract-based applications involving exchanging IoV data and cryptocurrencies. Furthermore, the paper conducts a thorough security analysis, evaluating the proposed framework against various attack scenarios, and presents comprehensive performance metrics from real-world implementation [39].

Existing data management solutions for IoV data encounter challenges in verifying data source credibility and maintaining data security [40]. However, the decentralized nature of blockchain technology offers promising solutions to address these challenges. Nonetheless, obstacles remain to overcome, such as computing resource consumption and data segmentation between on-chain and off-chain components. To solve these issues mentioned above, the paper presents a recent study proposing a blockchain-based information management system designed to handle vehicle-related data reliably [41]. The system adopts a sharding blockchain structure, where IoV nodes within a particular geographical area form sub-chains to enhance computational speed and reduce the workload on the main chain. Additionally, an on-chain and off-chain computing scheme based on trusted execution environments is proposed to ensure data source credibility and optimize the computational resource consumption of smart contracts [42].

In vehicle insurance, UBI has emerged as a contemporary and practical approach. Unlike traditional methodologies, UBI determines insurance premiums based on real-time driver behavior. However, challenges arise due to the need for more transparency in claims processing, resulting in delays and increased fraudulent activities. In response, decentralized technologies, particularly blockchain, have gained attention as a promising solution. Consequently, this study proposes a blockchain-based framework for implementing vehicular UBI and incentives within the IoV context.

## 3. Research Method

### 3.1. Design of Blockchain IoV System Infrastructure

The blockchain consensus mechanism aims to achieve decentralization, security, and scalability simultaneously [37]. However, increasing transmission speeds may entail compromising decentralization, while prioritizing decentralization may result in reduced scalability. Existing blockchain platforms face challenges in meeting all three objectives concurrently. For instance, delegated proof of stake (DPoS) sacrifices decentralization to achieve high transmission speeds [43]. By contrast, proof of work (PoW) maintains decentralization but struggles to attain high transaction speeds [44]. Tendermint, a consensus blockchain mechanism, addresses these challenges by introducing asynchronicity to the Ethereum blockchain [45]. Tendermint primarily focuses on enhancing fault tolerance and efficiency. To overcome these challenges, the whole Ethereum blockchain incorporates the essential functions of Ethereum by integrating the Tendermint consensus algorithm. Nodes within this system are categorized into externally owned accounts (EOAs) and smart contracts. The platform determines the voting rights of EOAs for financing purposes based on the stakeholders’ influence. Moreover, within the Tendermint system, every node assumes the role of a verifier, possessing the necessary privileges to access, transmit, and authenticate transactions. This characteristic aligns with the blockchain DNS solutions [46], which similarly adopts a decentralized gateway architecture to connect private blockchains with end-users. This is accomplished by harnessing the distinctive features of both public and private blockchain platforms through its underlying Ethereum structure [47].

This research platform is developed based on the architecture of Tendermint, which integrates a security auction mechanism to address apprehensions regarding potential data leaks arising from operational negligence. To ensure authentication for all auction nodes and participants and minimize data loss risk, a public–private key pair is assigned to each bidder. This research presents an architectural framework for the UBI blockchain policy auction mechanism in IoV applications. It includes a customized auction mechanism, a decentralized bidding algorithm, and intelligent contract functionality to enhance the security, fairness, and trustworthiness of the system. The experimental evaluation validates the efficiency of the proposed algorithm, highlighting its potential for real-world implementation in the UBI domain.

In an IoV system, driving data are standardized and preprocessed before being uploaded to the blockchain platform (Figure 1). The IoV sensor on vehicles collects driving habits and transmits them to the blockchain platform. The platform identifies the vehicle, assesses the driver’s risk and driving habits, generates a risk judgment model, and adjusts the premium accordingly. The driving data collected are used to establish a risk evaluation model that results in a higher dividend, increasing the insurance company’s profitability. Onboard sensors record the behavior of a driver whenever they operate a vehicle, and this behavior is integrated, analyzed, encrypted, and subsequently uploaded to the UBI platform based on blockchain technology. Following this, the data undergo several validation and confirmation processes performed by dealers, manufacturers, and maintenance depots. After verification, insurance agencies can access this data to evaluate the driver’s behavior, modify the premium, and create a new tailored contract that includes different insurance products for the driver’s selection and also provide driving advice. This approach enables insurance pricing to be based on two risk factors: driver-related and vehicle-related. Consequently, there are three types of insurance plans: (1) billing the driver and vehicle owner separately based on their risks; (2) billing only the vehicle owner based on both driver-related and vehicle-related risks; and (3) billing only the driver based on driver-related risk.

### 3.2. Auction Mechanism

This study constructs a driver blockchain that is based on the driver and vehicle information collected by an IoV-capable vehicle system that meets the insurance requirements of the vehicle owner. After the driver blockchain is completed, the blockchain-based UBI platform generates an auction agent and connects to an insurance company’s auction platform. The driver’s insurance requirements are stored on the auction platform. After auction on the auction platform, the auction agent automatically matches the bid with the most appropriate insurance portfolio. The detailed process is illustrated in Figure 2. The primary participants in the designed blockchain-based UBI platform are as follows:(1)Broker blockchain pairs bidders and auctioneers, generates auction lists, and gives bid price information to users.(2)Auctioneer blockchain generator is used by individuals to participate in auctions and place bids.(3)Driver blockchain collects driving habits and UBI requests.(4)Broker blockchain pairs auctioneers, generates auction lists, and gives bid price information to users.(5)Auction server operation is a Web interface offered by the auction house server that enables auctioneers to execute the set of instructions and has two parts: First, to specify the auctioneer agent, the auctioneer sends a request and auction information to the auctioneer agent generator, which creates an agent from a pre-existing template. The newly created agent and auction information are then registered with the broker agent. Second, the auctioneer can control the auctioneer agent’s behavior by communicating with it directly.

**Figure 2 sensors-23-06482-f002:**
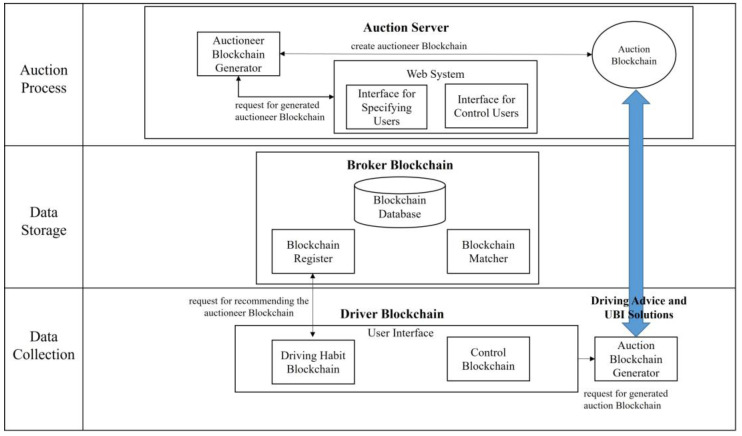
System structure of the blockchain-based UBI platform with auction algorithm.

## 4. System Implementation

### 4.1. Blockchain-Based Auction Algorithm for UBI

Elliptic curve cryptography and a first-price sealed-bid auction was implemented to develop a suitable auction mechanism for the UBI blockchain platform [48,49,50]. The proposed blockchain auction mechanism has four phases: auction initialization, bidder registration, auction process, and winner announcement. This process has four main participants: bidder registration manager, bidder agent house, blockchain auction house, and bidder.

(1)Bidder registration manager oversees the mandatory bidder registration process prior to the commencement of an auction, allowing bidders to participate in multiple auctions following registration. Additionally, the manager stores and verifies the corresponding secret parameters of registered bidders. They also manage and maintain a virtual bulletin board containing two types of information: a bidder’s registration key and identity information, as well as their pseudonym utilized in a single auction round. Access to the board is exclusive to the bidder registration manager, who has the ability to write and update it, while all users may refer to the posted information for identity verification purposes.(2)Bidder agent house is an entity that is responsible for establishing communication with the broker blockchain and creating bid agents. Its primary function is to manage and maintain a bulletin board that stores the public keys of bidders’ transactions, which can be used for verification purposes.(3)Blockchain auction house provides the auction location, hosts auctions, and manages auction operations. It also maintains a virtual bulletin board that contains the auction information of all bidders and the winning bidder for verifying bidder identities.(4)Bidder is an insurance company that participate and places bids in the auction.

### 4.2. Auction Algorithm Initialization

The parameters for the auction algorithm are listed in Table 1.

The bidder registration manager and bidder agent house establish system parameters through the following steps:(1)Bidder Registration Manager

Step 1: Set up a read-only virtual bulletin board and post all bidders’ registration keys, identity information, and pseudonyms used by the bidders in the *j^th^* auction round. Only the bidder registration manager can write to and update the bulletin board.

Step 2: Select a prime number greater than one million as *p*.

Step 3: Develop an elliptic curve equation where a and b satisfy $4a^{3}+27b^{2}\mathrm{mod}p\neq 0$

Step 4: Declare *G* and *Q*.

(2)Bidder Agent House

Step 1: Set up a read-only virtual bulletin board and post the information of all bidders and the transaction public key on the board. Only the bidder agent house can write to and update the bulletin board.

Step 2: Select *S_AH_* randomly to calculate a *P_AH_* that satisfies:*P_AH_* = *G*∗*S_AH_*
(1)

Step 3: Post *P_AH_* on the read-only virtual bulletin board.

(3)Registration

The bidder registration manager generates a pseudonym for a bidder before it joins a new auction and also encrypts the information into the hash function. The pseudonym can only be used while the auction is in progress, and the bidders will obtain their private key. The registration process is described as follows:

Step 1: Select *S_Bi_* as the private key of bidder *i* and calculate the corresponding registration key *R_Bi_* as follows:*R_Bi_* = *G*∗*S_Bi_*(2)

Step 2: Each bidder selects a secret parameter *S_i_*.

Step 3: Each bidder select a random integer *B*_1*,i*_ and calculates the verification information (*Ψ_i_*, *δ_i_*) as follows:*S_i_* = *G*∗*B*_1*,i*_ = (*xF*_1*,i*_, *yF*_1*,i*_) (3)
*Ψ_i_* = *H*(*xF*_1,*i*_|| *yF*_1,*i*_) (4)
*δ_i_* = *B*_1*,i*_ + (*Ψ_i_*∗*S_Bi_*) mod q (5)

Step 4: Each bidder transmits *R_Bi_*, *S_i_*, *Ψ_i_*, and *δ_i_* to the bidder registration manager.

Step 5: The bidder registration manager begins the registration progress and verifies the information by using the following equations:*S_i_*′ = *G*∗*Ψ_i_* − *R_Bi_*∗*δ_i_* = (*xF*_1,*i*_′, *yF*_1,*i*_′) (6)
*Ψ_i_*′ = *H*(*xF*_1,*i*_′|| *yF*_1,*i*_′) (7)
*Ψ_i_*′ ≟ *Ψ_i_*
(8)

Step 6: The bidder registration manager posts the registration information *R_Bi_* of bidder *i* on the virtual bulletin board.

Step 7: The bidder registration manager generates a pseudonym for each bidder before the *j^th^* auction begins in a random order.
*N_ij_* = *H*(*S_i_*)∗*R_Bi_*
(9)

Step 8: By using Equation (9), each bidder can obtain and verify its own pseudonym.

(4)Auction Progress

Step 1: The auction information for a bidder (*PK_ij_*, *B_ij_*) is sent to the bidder agent house to request a bid agent.

Step 2: The bidder agent house sends the auction information to the blockchain auction house, and *PK_ij_* is verified using the virtual bulletin board.

Step 3: The blockchain auction house announces the auction information of *B* on the virtual bulletin board.

(5)Winner Announcement

The bidder who places the highest bid is the winner after the *j*th auction round, and the blockchain auction house verifies the authenticity of *B* before announcing the winner on the virtual bulletin board as follows:

Step 1: The blockchain auction house asks for the winner’s *PK_ij_*, *R_Bi_*, and *N_ij_* from the bidder agent house.

Step 2: The blockchain auction house checks the consistency of the *PK_ij_*, *R_Bi_*, and *N_ij_* by using Equation (9).

Step 3: After announcing the winner, anyone can verify the auction process by using the virtual bulletin board.

### 4.3. Security and Practicality Analysis

#### 4.3.1. Security Analysis

The following common security issues are discussed to analyze the security of the elliptic curve cryptosystem-based bidding mechanism applied to the IoV blockchain platform proposed in this study:(1)Anonymity: The bidder’s identity in the auction process is known only to the broker blockchain and auction server. It is not accessible to external entities.(2)Fairness: All participants in the auction use pseudonyms. The bidder agent house is responsible for sharing valid auction information on a virtual bulletin board. If a bidder cannot find their auction information, they can appeal to the bidder agent house. This ensures impartial administration of bidder information.(3)Efficiency and easy revocation of auction: The use of the elliptic curve cryptosystem reduces the computational burden of online auction operations. The broker blockchain can quickly remove a bidder’s identification and confidential parameters from the database. Once this information is removed, the bidder loses their eligibility for future auctions.

#### 4.3.2. Practicality Analysis

A comparative analysis of the merits of utilizing blockchain technology in the proposed insurance policy auction mechanism reveals significant advantages compared to a system without blockchain were discussed as follows:

In a traditional system lacking blockchain integration, the insurance policy auction mechanism operates under centralized control, typically governed by a central authority or intermediary. Such centralization introduces potential bottlenecks, delays, and increased costs due to the reliance on intermediaries [51]. Additionally, storing sensitive customer data on centralized servers exposes it to vulnerabilities, including security breaches and unauthorized access, thus jeopardizing drivers’ information and privacy. Consequently, the policy issuance and settlement processes must be more active and effective [49].

The adoption of blockchain technology in the proposed insurance policy auction mechanism offers notable advantages. Blockchain’s inherent decentralization ensures a distributed auction mechanism, rendering intermediaries unnecessary. Participants gain equal access to an immutable and transparent ledger, establishing transparency and fairness within the auction process. This decentralized nature also fosters trust and verifiability among participants, as reliance on intermediaries is replaced by trust in blockchain technology’s protocol and consensus mechanism [50]. Blockchain’s implementation guarantees enhanced security through cryptographic algorithms and a distributed ledger system. Encrypted and distributed data storage across multiple nodes diminishes the risk of data breaches and unauthorized access, thus safeguarding drivers’ sensitive information. By eliminating the need for manual interventions, smart contracts streamline operations, reducing administrative costs and processing times. Furthermore, blockchain technology facilitates efficient settlements, significantly reducing settlement times [51]. It becomes evident that incorporating blockchain technology in the proposed insurance policy auction mechanism provides substantial merits. These include decentralization, transparency, enhanced security, automation, efficient settlements, improved data management, and global accessibility. Collectively, these advantages contribute to establishing trust, fairness, efficiency, and cost reduction, thereby benefiting insurers and drivers.

### 4.4. System Simulation Experiment

In this section, system simulation experiment for a blockchain-based UBI policy auction mechanism in the IoV is developed to evaluate the feasibility and performance. By utilizing a simulation model that incorporates the essential components of blockchain technology and the IoV environment, this experiment seeks to assess the effectiveness of the UBI policy auction mechanism. Through the design of various scenarios, parameters, and performance metrics, the simulation experiments can provide valuable insights into the efficiency of the blockchain-based UBI policy auction mechanism.

#### 4.4.1. Generate Time of the Auction When Driving Behavior Changed

This experiment aims to assess the efficiency of a blockchain-based UBI auction mechanism in determining insurance premiums by leveraging real-time driving behavior data obtained from connected vehicles. Additionally, the study aims to evaluate the influence of vehicle-to-platform communication on a policyholder’s behavior and analyze the system’s scalability. This experiment was explicitly designed to examine the system’s operational speed under varying conditions of sensor activation, representing shifts in driving behavior. By investigating the relationship between sensor triggers and the resulting system performance, this research seeks to verify the observed trends and draw conclusions regarding the system’s responsiveness to changes in driving patterns. The results were shown in Figure 3.

In light of each driving event, the system must modify the insurance record per the particular event. Based on the empirical findings, it becomes evident that when the occurrence of driving events exhibits a significant rise, albeit with a slight increase in the system’s processing time for generating an insurance plan, it can be observed that the incremental increments are minute in nature. This observation underscores the system’s ability to effectively and promptly update the driver’s status information in real time.

#### 4.4.2. Different Number of Insurance Company

The number of bidders participating in an auction process significantly influences the operational speed of the overall system. The presence of a larger pool of bidders leads to an increased number of bidding rounds, complexity of the hash function, and a subsequent expansion of the auction’s scale. In the scope of this experimental investigation, as shown in Figure 4, we aim to empirically validate the operational efficiency of the proposed elliptic curve cryptography-based first-price sealed-bid auction algorithm on a blockchain platform, particularly when confronted with a scenario involving multiple insurance companies and an augmented number of stakeholders in the system.

The objective of this experimental study is to evaluate the efficacy of the system by augmenting the participant pool in the bidding process. Specifically, we investigate the duration required for the system to identify the winning bidder in scenarios where varying numbers of insurance companies submit bids. The experimental results reveal that as the number of insurance companies increases exponentially, the temporal overhead incurred by the system exhibits only marginal growth. Moreover, the allocation of bids is accomplished within a significantly abbreviated timeframe. Consequently, these findings substantiate the efficiency of the proposed algorithm.

## 5. Conclusions

### 5.1. Lesssons Learned

Individual driving behavior complicates the relationship between responsibility and obligation in traffic behavior. This study investigated driving insurance. In practice, the two key aspects of insurance plans are how insurance companies can learn about driving habits to evaluate insurance premiums and how consumers can protect their rights. These processes have various limitations and incur informational asymmetry. Previously, only gender and age were used to evaluate driving risk; this method is unsuitable in practice.

In conclusion, this research presents a robust framework for implementing a fair auction mechanism on a practical blockchain platform in the UBI and the IoV domains. Incorporating driving behavior recordings, auction mechanisms, and safety reminders contribute significantly to advancing these fields. Leveraging the capabilities of blockchain technology and IoV advancements enhances the platform’s overall efficiency, fairness, and trustworthiness, thereby establishing a mutually beneficial environment for drivers and insurance companies. The key contributions of this study can be summarized as follows:

Firstly, constructing the UBI system on the blockchain platform, combined with utilizing IoV technology, facilitates detailed recordings of driving behavior. By incentivizing drivers to adopt safe habits through reduced insurance premiums, this system promotes the cultivation of safer driving practices. Integrating IoV technology enables the comprehensive collection and analysis of driving data, providing valuable insights into driver behavior. Secondly, an auction mechanism has been designed for implementation on a blockchain platform utilizing elliptic curve cryptography and first-price sealed-bid auctions. These sophisticated technologies ensure both information security and auction fairness. By simplifying the process for drivers, who are only required to provide their insurance requirements, and enabling insurance companies to bid based on the drivers’ recorded behavior on the blockchain, this auction mechanism prevents drivers from being subjected to excessive charges. Simultaneously, it facilitates insurance companies’ understanding of drivers’ behavior, fostering a mutually beneficial environment. Thirdly, the development of a blockchain-based UBI maintenance system allows for the comparison of driving data and insurance claims, enabling the identification of correlations between various driving behaviors and associated risks. This knowledge facilitates the recommendation of insurance plans that align with a driver’s typical driving habits, ultimately enhancing the overall efficiency and effectiveness of the UBI system.

Additionally, implementing an auction mechanism appropriate for the blockchain-based UBI platform can address the issue of information asymmetry between insurance companies and consumers. By increasing transparency and fairness in the auction process, consumer trust and confidence in the platform can be enhanced, resulting in a more reliable and trustworthy UBI system. Integrating safety reminders into the system provides real-time notifications to drivers about the risks associated with different road sections during their journeys. Moreover, the system proactively recommends safer road sections through GPS navigation, thereby contributing to improved driver safety. The driving behavior model, based on the recorded data of vehicle owners on the blockchain, serves as a foundation for developing various application scenarios. Leveraging the communication capabilities of the IoV, novel forms of assistance can be achieved. For example, the system can promptly notify a nearby garage in case of an abnormal tire pressure sensor reading, prompting the driver to inspect and repair the tire to prevent potential accidents.

### 5.2. Future Work

This research showcases a comprehensive framework for implementing a fair auction mechanism on a practical blockchain platform within the UBI and IoV domains, integrating driving behavior recordings, auction mechanisms, and safety reminders. By harnessing the potential of blockchain technology and IoV advancements, the platform achieves improved efficiency, fairness, and trustworthiness, fostering a mutually beneficial environment for drivers and insurance companies.

Future work in this area should address the complexities of individual driving behavior and its implications for the relationship between responsibility and obligation in traffic behavior. While this study has specifically investigated driving insurance, two key aspects require further attention: how insurance companies can effectively learn about driving habits to accurately evaluate insurance premiums and how consumers can protect their rights. Current approaches have limitations and often result in informational asymmetry. Previous methods that relied solely on factors, such as gender and age to assess driving risk are inadequate in practice. Future research can explore innovative approaches to gathering and analyzing driving behavior data to advance this field, which could involve developing more sophisticated technologies and methodologies for evaluating driving habits and determining insurance premiums. Insurance companies can gain deeper insights into individual driving behavior and accurately assess risk levels by leveraging emerging technologies and techniques, such as advanced data analytics, machine learning algorithms, and artificial intelligence. This would contribute to fairer and more personalized insurance plans.

## Figures and Tables

**Figure 1 sensors-23-06482-f001:**
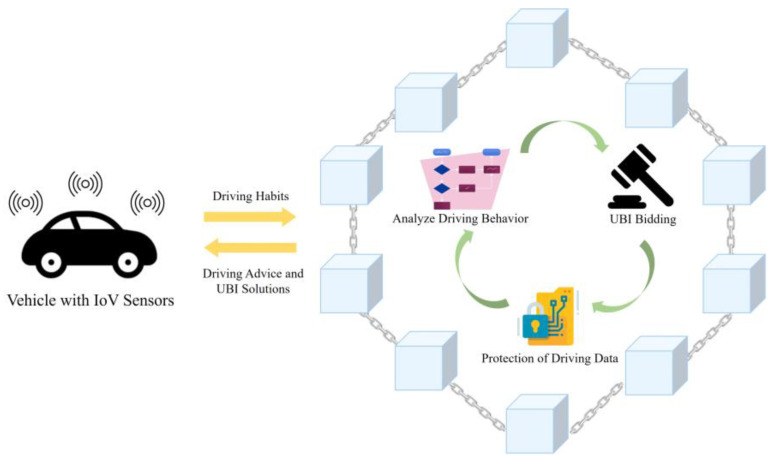
Blockchain-based UBI system model.

**Figure 3 sensors-23-06482-f003:**
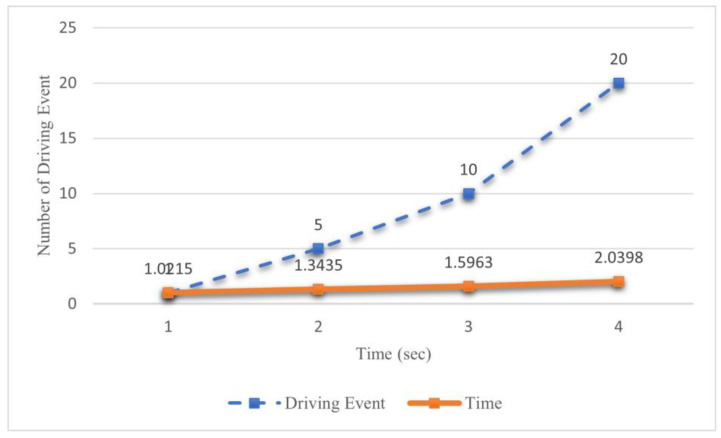
Experiment results when a different number of driving events happens.

**Figure 4 sensors-23-06482-f004:**
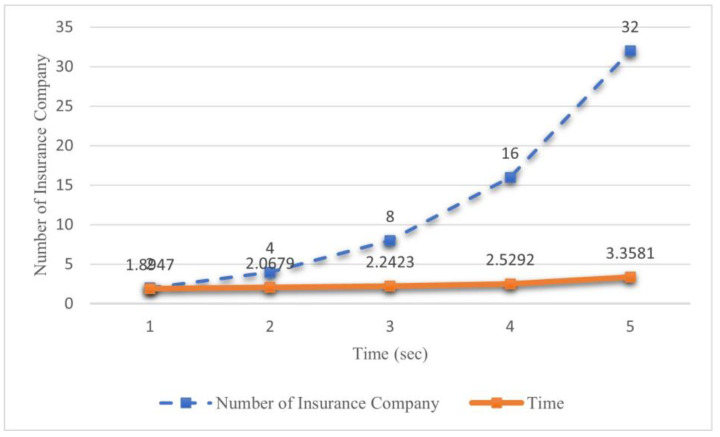
Experiment results in different number of insurance company.

**Table 1 sensors-23-06482-t001:** Parameters for the Auction Algorithm.

Parameter	Description
*B*	Set of all bidders in the auction {1, 2, 3, …, *i*}
*P*	Set of large prime number includes {1,2,3…*p*}
*Q*	Set of large prime number includes {1,2,3…*q*} as the order of a generative point on the elliptic curve with value within $p+1\pm \sqrt[{2}]{p}$
*E*	The elliptic curve equation $y^{2}=x^{3}+ax+b(\mathrm{mod}p)$where *a* and *b* satisfy $4a^{3}+27b^{2}\mathrm{mod}p\neq 0$
*G*	A point on the elliptic curve
*F*	A point on the elliptic curve
*xF*	The x-coordinate value of *F*
*yF*	The y-coordinate value of *F*
*H*(*x*)	A hash function that satisfies *H_i_*(*x*) = *H* (*x*, *H_x−_*_1_(*x*)) and *H_o_*(*x*) = *x*
*S_AH_*	The private key of the bidder agent house
*P_AH_*	The public key of the bidder agent house
*S_Bi_*	The private key of *B_i_*
*R_Bi_*	The registration key of *B_i_*
*B_ij_*	The price of Bi in the *j^th^* auction round
*S_i_, B* _1*,i*_ *, B* _2*,i*_	Three secret parameters chosen by Bi
*N_ij_*	A pseudonym created by the bidder registration manager for *B_ij_*
*R_j_*	Random number chosen by the bidder agent house in the *j^th^* auction round
*P_Ij_*	The public information provided by the bidder agent house in the *j^th^* auction round
*PK_ij_*	The transaction public key generated for *B_ij_*

## Data Availability

Not applicable.

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
