# Peer review of "A Trustable and Secure Usage-Based Insurance Policy Auction Mechanism and Platform Using Blockchain and Smart Contract Technologies"

_sensors, 2023, doi:10.3390/s23146482_

Round 1

Reviewer 1 Report (Previous Reviewer 1)

1, The results section should show the merits of using blockchain by comparing your Insurance Policy Auction Mechanism with and without blockchain. You can refer to some examples including doi: 10.1038/s41560-022-01027-4.

2, Some features in Section 4.3 are common knowledge for any application on blockchain. Please simplify them. 

3, Which blockchain platforms do you use to develop the auction?

English is OK

Author Response

Please see the attatched file.

Reviewer 2 Report (Previous Reviewer 3)

All problems have been solved.

Author Response

We sincerely thank the reviewer for the affirmation!

Reviewer 3 Report (Previous Reviewer 2)

The authors have responded to our questions and updated them. The manuscript can then be accepted for further processing in order to be published.

Author Response

We sincerely thank the reviewer for the affirmation!

Round 2

Reviewer 1 Report (Previous Reviewer 1)

My comments are addressed. 

acceptable

Author Response

We sincerely thank the reviewer for the affirmation!

This manuscript is a resubmission of an earlier submission. The following is a list of the peer review reports and author responses from that submission.

Round 1

Reviewer 1 Report

1, The contributions are somehow scattered. Please clearly show the relations of each of your contributions.

2, The authors are encouraged to show numercially, not just by words, the advantages of using blockchain for IoV.  Examples include doi: 10.1109/TPWRS.2021.3086101, doi: 10.1038/s41560-022-01027-4, and doi: 10.1016/j.adapen.2021.100029.

3, There have been many blockchains in literature. How do you tailor existing blockchains to fit your specific Insurance Policy Auction and Internet of Vehicles (IoV) applications? 

English can be improved. 

Reviewer 2 Report

Greetings,

Very intresting paper, however following are my observation and i request authors to attend them.

Abstrat is not clearly written hence it can be improved further. We would like to see some results/comparission with state of the art.

Introudction :contains some not relevent information with respect UBI, so delete and your MS focus must be on UBI itslef.

Literature is having some issues

 IoV Infrastructure With Blockchain:-- very old references considered, improve find more  recent references.

1 Design of Blockchain IoV System Infrastructure

If Figure 1 is this author's contribution, please describe or otherwise elaborate upon what exactly their contribution is.

3.2 Auction Mechanism: Timing diagram will be better in this and describe the flow of the same.

4. System Implementation

4.1 IoV and Blockchain-based UBI Platform

Following this, the data undergoes several 254 validation and confirmation processes performed by dealers,? what exactly and how it goes on perfroming such complitcated issues/

How it is implemented? simulatd? please more information on this needed.

Testing? simulation ?resutls? needed?

Reviewer 3 Report

1. What is the main question addressed by the research?

This research presents an architectural framework for the Usage-Based Insurance (UBI) blockchain policy auction mechanism in Internet of Vehicles (IoV) applications. This study aims to analyze and design the blockchain architecture and management considerations specific to the UBI environment.

2. Do you consider the topic original or relevant in the field? Does it address a specific gap in the field?

Yes. The problem solved in this paper is relevant to this journal.

3. What does it add to the subject area compared with other published material?

The main contributions of this paper include:

- The UBI system is constructed on the blockchain platform and uses IoV technology to record detailed driving behavior to encourage drivers to acquire safe habits and thus reduce their insurance premiums.

- An auction mechanism is established that can be implemented on a blockchain platform with elliptic curve cryptography and first-price sealed-bid auctions.

4. What specific improvements should the authors consider regarding the methodology? What further controls should be considered?

- What is the motivation of using blockchain as the building block to design the method?

- In the proposed system model, more details about the data stored and transmitted on blockchain could be provided.

- How to combine the authors’ work with these works in terms of blockchain data sharing and queries, such as blockshare: a blockchain empowered system for privacy-preserving verifiable data sharing, vql: efficient and verifiable cloud query services for blockchain systems, vchain+: optimizing verifiable blockchain boolean range queries. More discussion should be added in the paper.

- There are more opportunities for conducting meaningful experiments to comprehensively evaluate the system performance and overheads. Some experimental results could be provided.

- Some potential research directions and future works could be discussed in this paper.

5. Are the conclusions consistent with the evidence and arguments presented and do they address the main question posed?

Yes.

6. Are the references appropriate?

More technical papers about blockchain should be investigated and analyzed. For example:

- Leveraging public-private blockchain interoperability for closed consortium interfacing. IEEE Conference on Computer Communications.

- VFChain: Enabling verifiable and auditable federated learning via blockchain systems. IEEE Transactions on Network Science and Engineering.

- SymmeProof: Compact Zero-Knowledge Argument for Blockchain Confidential Transactions, IEEE Transactions on Dependable and Secure Computing.

- Unearthing malicious campaigns and actors from the blockchain DNS ecosystem, Computer Communications.

7. Please include any additional comments on the tables and figures.

- Tables and figures are easy to understand.